# Measuring information flux between social media and stock prices with Transfer Entropy

Román Alejandro Mendoza Urdiales[1][ID][☯]*, Andrés García-Medina[2,3][ID][☯], José Antonio Nuñez Mora[1]

**1** Department of Finance, Tecnológico de Monterrey, EGADE Business School, Ciudad de México, Alvaro Obregon, México, **2** Consejo Nacional de Ciencia y Tecnología, Ciudad de México, México, **3** Unidad Monterrey, Centro de Investigación en Matemáticas, PIIT, Apodaca, Nuevo León, México

☯ These authors contributed equally to this work.
* A00939178@itesm.mx

**Data Availability Statement:** All relevant data are within the paper and its Supporting information files.

**Funding:** FOSEC SEP-INVESTIGACIÓN BÁSICA through Consejo Nacional de Ciencia y Tecnología,

## Abstract

Transfer Entropy was applied to analyze the correlations and flow of information between 200,500 tweets and 23 of the largest capitalized companies during 6 years along the period 2013-2018. The set of tweets were obtained applying a text mining algorithm and classified according to daily date and company mentioned. We proposed the construction of a Sentiment Index applying a Natural Processing Language algorithm and structuring the sentiment polarity for each data set. Bootstrapped Simulations of Transfer Entropy were performed between stock prices and Sentiment Indexes. The results of the Transfer Entropy simulations show a clear information flux between general public opinion and companies' stock prices. There is a considerable amount of information flowing from general opinion to stock prices, even between different Sentiment Indexes. Our results suggest a deep relationship between general public opinion and stock prices. This is important for trading strategies and the information release policies for each company.

## Introduction

There is a large strand of the finance literature interested in demonstrating how the news related to a publicly-traded company affect its stock price performance, and how there is a direct correlation between the direction and magnitude of the stock price response and the nature of the published news [1].

Several studies conclude that the news' impact is related to the media coverage of the company [2]. Furthermore, still more studies propose that whether the information is public or private is not relevant and that what matters is that traders have access to it [3]. Eugene Fama stated that the information available about traded companies is fully reflected in the prices, naming this theory the Efficient Market Hypothesis [4].

The Efficient Market Hypothesis (EMH) has been challenged by evidence in diverse studies in different markets and periods of time. Different phenomena not explained by the EMH can be justified from Adaptive Market Hypothesis (AMH) and the Fractal Market Hypothesis (FMH). Studies like Kim, Lim and Shamsuddin [5] for the US data, Shi, Jiang, Zhou [6] for the

México, A1-S-43514 (https://conacyt.mx/ciencia-de-frontera/, AGM). The funders had no role in study design, data collection and analysis, decision to publish, or preparation of the manuscript.

**Competing interests:** Enter: The authors have declared that no competing interests exist.

Chinese market, Árendáš and Chovancová [7] for the very known group of Brazil, Russia, India and China, studied predictability and concluded consistency with the AMH. In the case of the Nigerian stock market, Adaramola and Obisesan [8] found out linear and non-linear predictability and unpredictability periods, i.e., they establish that this market is not efficient and follows the concept of Adaptive Market Hypothesis. For the Vietnamese stock market, D Phan Tran Trung et. al [9] founded that the empirical evidence supports the AMH. A similar result is discovered for the Dhaka Stock Exchange studying seasonal anomalies of the market in Akhter and Yong [10]. For the case of investment we can study the market for example in periods of turbulence, see for example Moradi et. al [11] where the FMH is confirmed for the Tehran stock exchange and not confirmed for the London stock exchange. In Kristoufek L. [12] the FMH is accepted because of the short term investments dominance over long term investments in the case of financial turmoils. This study is developed for developed markets NASDAQ, FTSE 100, DAX,CAC,HIS and NIKKEI. In the same line Dar et. al [13] studied a long period of time (since mid-eighties) including important events like Black Monday (1987), subprime crisis (2008) and dotcom (2000) and again as in Kristoufek L. [12] the FMH is confirmed.

Farag and Cressy [14] studied how price limits, aimed to prevent speculation amongst traders when new information is released in the market. In their study, Farag and Cressy, extend Fama's EMH in two trends. The first one (Mixture of Distribution Hypothesis; MDH), assumes that the information in the market is available to all traders simultaneously and in consequence, the market achieves equilibrium immediately. The second hypothesis (Sequential Information Arrival Hypothesis SIAH), states that the investors receive information sequentially and in a random manner and the market is adjusted by the traders according to when the information is received over time. Farag and Cressy concluded that there is inefficiency in the information dispersion and that volatility of the stock market is increased instead of reduced when price limits are regulating the market.

There are studies that compare trading regulations between countries [15] and present the differences between laws and penalties regarding similar crimes in different countries. These studies trace back the regulation of securities markets as far as the beginning of the century. Other studies [16, 17], remark gaps in securities regulation considering that interconnected institutions are affected by the same issues, even more perceived during international financial crises. And how regulators are making an effort to address the differences between countries and standardize them, especially ones considering market transparency and data consolidation.

There are some documented cases, in which a negative false announcement referred to a given company affected its stock prices heavily during a short period of time, but when the news was revealed to be fake, the stock price did not recover entirely. An example happened in September 2008, when United Airlines stock dropped more than 75% due to a six-year-old article that resurfaced on the Internet about the 2002 bankruptcy of the United's parent company, mistakenly believed to be reporting a new bankruptcy filing. However, once the news was cleared, the stock price still ended 11.2% below its prior valuation [18]. The United Airlines case illustrates the effect of high volatility after a news release called "drift". The drift is usually present after the news release and its amplitude depends on the nature of the news. Evidence suggests that companies with negative news releases have longer drifts than companies with positive news [19].

Numerous publications aim to build models to predict stock prices, considering the traditional models have not been fully successful in doing so. In contemporary markets, stockholders' opinions are considered faithful indicators of the future value of their investment holding [20]. With the common use of social networks, the opinion of the stockholders has been more present than ever before. Social networks flux of opinions, in combination with the traditional prediction models, have improved the rate of success of prediction methods significantly.

There are several published studies that report new models to predict stock prices using mostly social media opinions [21]. While some studies achieve some degree of prediction capability [22, 23] some others conclude that social sentiment is not useful for stock price prediction [24, 25]. Sentiment analysis of social media has also been used to study the effect of news releases on the price of cryptocurrencies [26]. It can even be stated that cryptocurrency prices are more susceptible to volatility because this category of assets has not yet gained the complete trust of investors.

There is a study that aims to measure how publicly available macroeconomic news influence stock returns by applying Auto-Regressive Vectors (VAR) [27]. The study concludes that there definitely are market movements that coincide with major economic events. The author states that there is also information not considered in the study that affects the stock price performance. Another work, while using Ordinary Least Squares (OLS) regressions, present how the percentage of negative words of news mentioning the situation of a particular firm has a direct impact on the stock return [28]. The study states that the negative words in firms' news precede low earnings and that in consequence a delayed impact on the stock price. The volume of information analyzed varies greatly between both studies, largely to the technology available at the time of each research, and thus the software and methodology applied to analyze the information. Both authors mention this issue, the first study, released in 1988, remarks that in their analysis they failed to consider more information sources to correctly account for the price volatility whether in the second study the authors emphasize the importance of considering a complete set of events is crucial in identifying patterns in firm responses and market reactions to the events. A different study [2], presents a linear regression model to describe the impact of information flux from news on trading activity in Nokia stock price. The results present a dependency on both volatility of the stock price and news regarding the company. The authors remark the relationship of news sentiment in Nokia stock returns.

In this study we compare how the information flux has been evolving through time. Since social media phenomena are recent, it was not considered in early studies regarding the news impact on stock prices [27]. Two conclusions can be drawn comparing our model with early work; first, there is definitely an information flux between the stock market and news media. Second, depending on the window of time analyzed, the information signal could be still in the stock market or in the news media, meaning that there is a delay between the signal emission, the reception in the stock price, and when the signal can be measured in its largest magnitude. A major concern here is understanding how the signal flux is structured. Tetlock [28] states in his research paper that investors mostly get their information secondhand. Investors initially put more attention on the news/social media than directly in the company's reports or activities. We can infer that there is an initial signal sent from the stock market towards social media yet there is also influence from the news(directly and indirectly) regarding the firms' activity in the stock market in social media and, in consequence there is an impact on the stock market.

There has been well established that news mentioning public traded companies are considered a factor in stock returns performance [29]. But the question that current academia is trying to solve is, How can we translate this impact in a quantitative method and in a statistically measurable fashion that can be scientifically replicated?

Transfer Entropy(TE) is a recent method that measures the statistical coherence between systems through time. This method proposed by Thomas Schreiber [30] considers the exclusion of information that could affect the experiment results by irrelevant characteristics such as common history or input signals. This method usually applied for natural sciences has been lately applied in finance to understand better the causal relationships of exogenous variables in public traded companies [28, 31].

In this study, Transfer Entropy and other advanced computational techniques, web-scraping and text mining with Natural Language Processing (NLP) to build an index that measures the information flux from social media to prices and vice versa. In this way our Ex-ante expectations, which are proved to be consistent with the existing literature on the subject, are that there will be a positive information flux from social media to stock prices.

We analyze the general public opinion on Twitter ([www.Twitter.com](www.Twitter.com)) and its impact in the behavior of publicly-traded companies' stocks has been analyzed. The simulations show evidence that the information flux exists from public opinion towards the stock market. By combining two computing methods, web scraping and Natural Language Processing, and by constructing a time series we propose a method for measuring the impact of the news in the stock market behaviour giving room for predicting stock price returns monitoring the news of public traded companies.

The article is structured as follows. In the first section the variables used and the construction of the data indexes are declared. Section two explains the methodology to structure and preprocess the information. Section three presents the main results and compares them with existing methodology in prior literature. The fourth section presents the discussion under different perspectives. Finally, the last section states the conclusions and proposes further work.

## Materials and methods

For the construction of the sets, two main data sources were considered. The daily closing prices for the largest publicly traded companies in the world and operating in multiple markets (Nasdaq, NYSE, BCBA, BMV and OTC for the case of Tencent). This condition was followed under the premise that the larger the company the larger the public opinion information that would be available in social media. The second data set was obtained with help of a web scraping software that selected specific mentions in a determined period of time and language. The ticker for each company was searched for the tweets in the English language for the same period as the stock prices time series, from period 2013–2018. The filter selected for reading the information during the search was the top tweets. Each company index was constructed individually and the data sample obtained from Twitter varied widely, the company with the smallest sample of tweets was Royal Dutch (ticker $RDSA), with 459 mentions and a market capitalization of 227.61 billion U.S.D. as of August 23, 2019. The largest sample retrieved corresponded to Microsoft, with a current market capitalization of 1.018 Trillion U.S.D. The full list of companies analyzed in this study can be found in Table 1

## Methodology

In Fig 1 the three-phase framework applied for our model is presented. In which two A.I. robots were used. Robot 1 used for automated web scraping and text mining, Robot 2 was used for Natural Language Processing in which the tweets were filtered, prepossessed and the polarity was calculated. The analysis was performed after the data was analyzed and structured, pairing the index with the corresponding daily closing stock price.

### Phase 1

**Extraction with text mining.**   The technique used for retrieving the comments for the tickers of each company is known as web scraping. Web scraping is a data mining method in which information is retrieved from selected web pages to create large pools of information that may then be analyzed to find new patterns [32]. It is considered our first step for analyzing information.

Table 1. Characteristics of the analyzed companies.

| Companies Profile | | |
|---|---|---|
| Company | Country of Origin | Ticker |
| Amazon | U.S.A. | $AMZN |
| Facebook | U.S.A. | $FB |
| JP Morgan | U.S.A. | $JPM |
| Tesla | U.S.A. | $TESLA |
| IBM | U.S.A. | $IBM |
| Berkshire | U.S.A. | $BRKA |
| Exxon | U.S.A. | $XOM |
| Visa | U.S.A. | $V |
| Wells Fargo | U.S.A. | $WFCC |
| Royal Dutch | U.K Dutch | $RDSA |
| Ten Cent | China | $TCEHY |
| Vokswagen | Germany | $VW |
| AT&T | U.S.A. | $ATT |
| Intel | U.S.A. | $INTEL |
| Johnson & Johnson | U.S.A. | $JNJ |
| General Electric | U.S.A. | $GE |
| SAP | Germany | $SAP |
| Microsoft | U.S.A. | $MSFT |
| Ebay | U.S.A. | $EBAY |
| Google | U.S.A. | $GOOG |
| Bank of America | U.S.A. | $BAC |
| Procter & Gamble | U.S.A. | &PG |
| Cisco | U.S.A. | $CSCO |

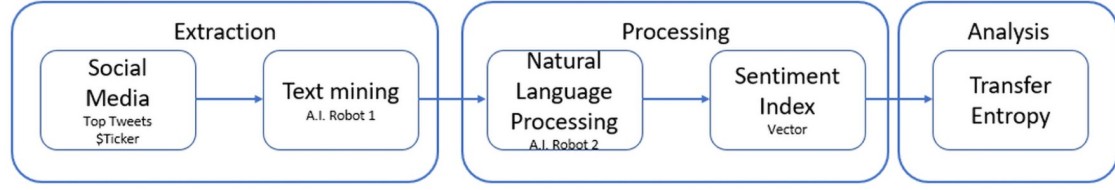

**Fig 1. Framework model in which we present the 3 processes developed in order to obtain, structure and analyze the information.**

For the text mining step, the social network was scanned with a JSON format written in Java, it is from open source and can be found online (https://webscraper.io/). The advantage of this robot (Robot 1) is that the information is extracted and structured in two columns, date and tweet. The text mining technique considered the following variables:

(a). Date: The period of time that was selected covered from February 1st 2013 until December 21st of 2018, which intended to cover most part of a full global economic cycle. The aftermath of the 2008 crisis until the preview of the economic deceleration of 2019.

(b). Language: The English was language selected for analyzing the information since it is the language chosen for business and most of the stock exchange is done in the US.

(c). Keywords: The only word that needed to be mentioned in each tweet was the company ticker(abbreviation used for trading preceded with the $ symbol)

(d). Top tweets: The page allows the search engine to classify the results in top tweets, a sample of 1% of the most commented and shared tweets.

The criteria were applied for the 23 companies, where each company was mined individually, meaning that there could possibly exist tweets that mention two or more companies. In that case, if Robot 1 detected the same tweet with two different tickers, this opinion was used in both Sentiment Indexes. In the final part, each unprocessed database was ordered chronologically. This gave room to compare data sizes and mention frequency since some analyzed companies began operating on stock market well after 2009 (Facebook IPO was in 2012).

### Phase 2

**Processing.** Natural Language Processing was applied for calculating the polarity in our unprocessed data vector for each comment extracted. For us to be able to construct our polarity vector each tweet was analyzed individually and the vector constructed posthumously.

Python was the language in which a Natural Language Processing algorithm was coded, and the specific library was TextBlob [33]. This library calculates Polarity by breaking each text analyzed individually into the words that compose the text. Single-letter words are ignored and for the rest of the text, a numeric value is given for each word that is already assigned inside the library, a value for polarity, subjectivity and intensity. When composed expressions are used (i.e. 'very great') the library considers interpretation rules that improve the analysis for structured sentences. While the total polarity is calculated with the simple addition of each individual polarity, the rules follow some structure, some of the rules are explained further:

1. One letter words are ignored

2. Negation words add the negative sign to the posterior word

3. Multipliers are words that emphasize the meaning of the next word.

In addition to the existing algorithm, the software was improved for the cleaning of each tweet phrase. The technique was changing abbreviations to the full extent of the words (i.e. 'ive', to 'I have', 'im' to 'I am', etc); this step was essential since the abbreviation of words is very common on Twitter given the limited space for each tweet (280 characters). By performing the additional cleaning for each tweet, the margin error for results with no value calculated was reduced considerably.

As an example, a real tweet from January 26, 2014:

```
When you have to set up an "ethics board" you know you don''t
have any.
Maybe time for @google to stop saying "don't be evil" $goog.
```

We can understand that the user is stating that the company Google lacks ethics in its directive board since it is setting up one and that the connotation of the sentence is not positive. The issue at hand is how to help the Robot to interpret the negativity as such. The first step is to clean the tweet from characters other than letters and from abbreviations. The algorithm returned the clean sentence:

```
when you have to set up an ethics board you know you do not
have any.
maybe time for google to stop saying do not be evil goog.
```

We can observe that all the characters that were not letters disappeared and the abbreviation *don't* extended to *do not*. The word *goog* since is not in the English dictionary was not cleaned with the library, unless you add it to its dictionary. For this particular case we let the word untreated.

In the second step the algorithm breaks the tweet into sentences and words:

- ```
  [Sentence("When you have to set up an ethics board you know you
  do not have any."), Sentence("Maybe time for @google to stop
  saying do not be evil $goog")]
  ```

- ```
  WordList(['When', 'you', 'have', 'to', 'set', 'up', 'an',
  'ethics', 'board', 'you', 'know', 'you', 'do', 'not', 'have',
  'any', 'Maybe', 'time', 'for', 'google', 'to', 'stop', 'say-
  ing', 'do', 'not', 'be', 'evil', 'goog'])
  ```

Finally, the algorithm analyzes the polarity and subjectivity for the sentence adding the individual score for each word. The individual score is also prerecorded in the library. Since the words *google* and *goog* are not in the library, they do not have a polarity value, meaning that the value assigned in the calculations will be zero not affecting the outcome of the analysis. The result for this example was a polarity $P = -1$ meaning that it has the highest negative score available.

It is important to mention that accuracy issues emerge when the words are scrambled (consider that this will definitely give a sentence with no sense) the calculated polarity will be different. For further reading review the source code [33].

**Sentiment Index vector construction.** Once the data sets have been processed and the polarity $P$ has been calculated for each tweet, the results were added for each observed day, it does not matter if the stock market was operating or not (including weekends and days off). For the day $t$ the Polarity value $Y$ was calculated with the simple addition of the Polarity $P$ of the $i$ tweets in the same day $t$:

$$Y_t = \sum_{n=1}^{i} P_n \tag{1}$$

$$Y = [Y_t, Y_{t+1}, Y_{t+2}, \ldots] \tag{2}$$

The variation of the stock $X$ for the day $t$, was calculated by subtracting the closing price from the day before($S_{t-1}$) to the closing price of the Stock of the day ($S_t$). In this manner vector $X$ is the stationary values of the prices:

$$X_t = S_t - S_{t-1} \tag{3}$$

$$X = [X_t, X_{t+1}, X_{t+2}, \ldots] \tag{4}$$

## Phase 3

**Transfer Entropy.** In this phase, the analysis of TE was applied to measure the flow of information between Sentiment Indexes vectors and stock market returns. The theory behind this methodology is explained further.

Let $Y$ and $X$ denote two discrete random variables with marginal probability distributions $p(x)$ and $p(y)$ with joint probability distributions $p(x, y)$, whose dynamical structures correspond to stationary Markov processes of order $k$ and $l$ for systems $Y$ and $X$ respectively. The Markov property considers the probability to observe $X$ at the time $t + 1$ in state $i$ conditional

on the $k$ previous observations is

$$p(x_{t+1}|x_t, \ldots, x_{t-k+1}) = p(x_{t+1}|x_t, \ldots, x_{t-k}), x_i \in X \tag{5}$$

Having $p(A|B)$ representing the conditional probability of A given B, $p(A|B) = p(A, B)/p(B)$, the TE from $Y$ to $X$ can be defined as the average information included in $Y$ excluding the information reflected by the past state of $X$ for the next state information $X$. Hence, TE measure is defined as [30]

$$T_{Y \to X}(k, l) = \sum_{i,j} p(x_{t+1}, x_t^{(k)}, y_t^{(l)}) \log \frac{p(x_{t+1}|x_t^{(k)}, y_t^{(l)})}{p(x_{t+1}|x_t^{(k)})}, \tag{6}$$

where $x_{t+1}$ of $X$ is affected by $k$ previous states of $X$, in other words, the lagged values affecting the current value of $X$. In addition, $X$ is affected by $l$ previous states of $Y$, in other words, the lagged values affecting the current value of $Y$.

TE attempts to incorporate time dependence into account by relating previous observations $x_i$ and $y_i$ in order to predict the next value $x_{i+1}$. Then, it quantifies the deviation from the generalized Markov property, $p(x_{i+1}|x_i, y_i) = p(x_{i+1}|x_i)$, where $p$ denotes the transition probability density to state $x_{i+1}$ given $x_i$ and even $y_i$. If there is no deviation from the generalized Markov property, $Y$ has no influence on $X$. Then, TE quantifies the incorrectness of this assumption and is formulated as the Kullback-Leibler entropy between $p(x_{i+1}|x_i, y_i)$ and $p(x_{i+1}|x_i)$ is explicitly nonsymmetric with respect to the exchange of $x_i$ and $y_i$.

An important property of TE is that under specific conditions it can be formulated as a nonlinear generalization of Granger causality. The last quantity plays an important role in the parameter estimation of a vector autoregressive (VAR) model in econometrics. There exists a series of results [34–36] that state an exact equivalence between the Granger causality and TE statistics for various approaches and assumptions on the data generating processes. It makes it possible to construct TE as a nonparametric approach of the Granger causality test. This relation can be regarded as a bridge between the information-theoretic approach and the causal inference under autoregressive models. It is important to mention that for highly nonlinear and non-Gaussian data, as is the case for most financial instruments, it is more adequate to model causality by using the TE information method instead of the traditional Granger causality test [37, 38].

On the other hand, the transfer entropy measure in Eq 6 is derived for discrete data. However, in any economic application the observed time series are continuous. There exist several techniques for estimating TE from observed data in order to apply it to real-world data problems. However, most of them require a large amount of data, and consequently, their results are commonly biased due to small-sample effects, which limits the use of TE in practical data. A straightforward approach to estimate TE is to partition the data into discretized values, for this particular case the data sampling was structured in. Thus, a time series $x(t)$ is partitioned to obtain the symbolically encoded sequence $S(t)$. This sequence replaces the value in the observed time series by the discrete states $\{1, 2, \ldots, n - 1, n\}$. The model allows to discretizing continue data by partitioning the data by choosing the quantiles of the empirical distribution of the time series. By denoting the bounds of the pre-selected number of bins by $q_1, q_2, q_3, \ldots q_n$, where $q_1 < q_2 < q_3, \ldots < q_n$. By performing this data partitioning, each value in the original time series is replaced by an integer.

Moreover, the expression of *TE* (Eq 6) is likely to be biased due to several factors as finite sample effects and the not strictly stationarity of financial data. Also, time series with higher entropy naturally transfer more entropy to the others. To reduce this bias, the Effective

Transfer Entropy (ETE) [39] has been proposed where

$$ETE_{J \to I}^{shuffled}(k, l) : T_{J \to I}(k, l) - T_{J_{shuffled} \to I}(k, l), \tag{7}$$

where $T_{J_{shuffled} \to I}$ indicates the transfer entropy from $J$ to $I$ with randomly shuffled time series $J$. Thus, all statistical dependencies between the *two-time series are destroyed. An important characteristic is that $T_{J_{shuffled} \to I}(k, l)$ converges to zero at a long sample size. Consequently, any non-zero value of $T_{J_{shuffled} \to I}(k, l)$ is due to small sample effects.

Later on, the work of Dimpfl et. al. [40] improves the bias correction by adding an inferences perspective of the estimated information flows. They proposed to use the Horowitz approach [41], which bootstrap the modelled Markov process. The idea is to simulate process $J$ based on the calculated transition probabilities, where the dependencies between $J$ and $I$ are destroyed, but the dynamics of the series $J$ is not changed. Transfer entropy is then estimated using the simulated time series. Then, this procedure is repeated several times to create a null distribution of no information flow, which can be used to test for statistical significance. The proposed equation has the same structure as ETE [40]

$$ETE_{J \to I}^{boot}(k, l) : T_{J \to I}(k, l) - T_{J_{boot} \to I}(k, l), \tag{8}$$

where $T_{J_{boot} \to I}$ indicates the average over the estimates derived from the null bootstrap distribution.

The null hypotesis p-value that measures if there is no information exchange is given by $1 - \hat{q}_{T_E}$, in which $\hat{q}_{T_E}$ denotes the quantile of the simulated distribution that corresponds to the transfer entropy estimations when the dependencies from $I$ to $J$ are destroyed.

## Results

Evidence of the influence of collective behaviour on stock price returns has been presented. This study provides an effective way to measure the relationship between social media and stock returns. Even so, it provides a potential way to measure the impact of social media in stock price performance. Taking the data sets one step further, we performed statistical analysis of the combination of all the vectors by the three methods (Pearson correlation, Granger causality and Effective Shannon Transfer Entropy).

In Fig 2 we compare the behaviour of the sock price returns and Sentiment Index of Facebook (ticker $FB). It can be observed that the activity in both vectors increases in the same periods of time. Short after its IPO in 2013, the stock movements and the activity in the Sentiment Index increases. The other period that can be observed with high movement is during

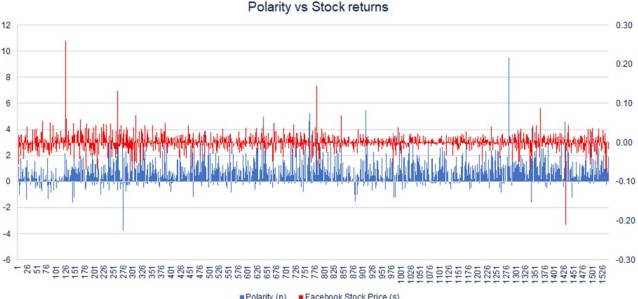

**Fig 2. Time series of daily returns of Facebook vs Facebooks' Sentiment Index.** It can visually be appreciated the synchronized upward or downward movements between the Sentiment Index and the company performance.

2018, where the company suffered largely negative publicity due to the Cambridge Analytica scandal.

The information flux between two data sets is usually measured with the correlation factor, which can be described as the statistical relationship between two variables. The most known method is the Pearson correlation coefficient which measures the linear sensitivity between two variables [42]. In financial analysis is not only important to know the level of the relationship between two variables, but the causal dependence between them. In Fig 3 we have the linear correlation matrix of the companies daily returns and the Sentiment Indexes(marked with * for each company) constructed for this research. The results show a clear positive correlation between companies daily returns and the indexes. Counter-intuitively, the matrix shows a little or no correlation between the companies and the indexes.

In order to measure the signal transfer (intensity and direction) between the Sentiment Index and the stock price individually, Shannon Transfer Entropy [43] was applied. The results support the theory of the effect of news media in asymmetric stock returns and the theory that the stock market performance is influenced by collective behaviour [44].

To analyze the information flux, we assigned the $X$ variable to the Stock Prices vector and the $Y$ variable to the Sentiment Index vector and the intensity of the signal was calculated:

$$\text{Intensity} = \text{ETE}(Y \to X)/[\text{ETE}(Y \to X) + \text{ETE}(X \to Y)] \tag{9}$$

With this expression, if the information signal was greater from $(Y \to X)$, it would take a positive sign, meaning that the information flows from social media to the stock market, which is the main interest of our study. The contrary sign could be expected if the signal from $(X \to Y)$ was greater, meaning that the stock market is affecting the activity of social media. This was performed exclusively to differentiate the direction of the signal in Fig 5.

We used the implementation RTransferEntropy to estimate the $ETE_{J \to I}^{boot}(k, l)$ considering the configuration $J = X$, $I = Y$, $k = 3$, $l = 3$, and a bootstrap simulations of 300 shuffles. The maximum p value considered was ($p \leq 0.10$) The quantiles selected to discretize the time series were c(5, 95), meaning that the 5% and 95% quantiles of the empirical distribution are used.

TE quantifies the information provided by the past of the process X influencing the present of the process Y, that is not already provided by the past of Y. With this implication, in this work we to quantify the information provided by the past of $Y$ on the shifted portion of the system $X$.

Simulations were performed considering lags = 1,2 and 3 for both variables X and Y, obtaining better results with k = l = 3. With these results, our work supports the theory that the sentiment signal is observable with more clarity in the stock price in a 3-day lag, considering that we are working with daily observations. This would mean that today's polarity will have its maximum effect on the stock price 3 days in the future.

For the construction of the $X$ vector(Eq 4), simple differentiation was considered, in order to work with the stationary expression of the stock price, as an alternate approach since most literature focus on working with the stock performance. The $X$ vector (Eq 4) was tested for unit root using two methods, Dickey-Fuller and Phillips-Perron. In every test, the results concluded that the $X$ vector was stationary. The same process was performed in the $Y$ vector (Eq 1) obtaining the same results. This to ensure that the model was executed with stationary vectors.

The first results are presented in Table 2. We can observe that there is a clear effect of the stock price performance given the Sentiment Index in TE simulations (p value = < 0.10) towards the stock price ($Y \to$ X). An important remark is that these twelve companies are from

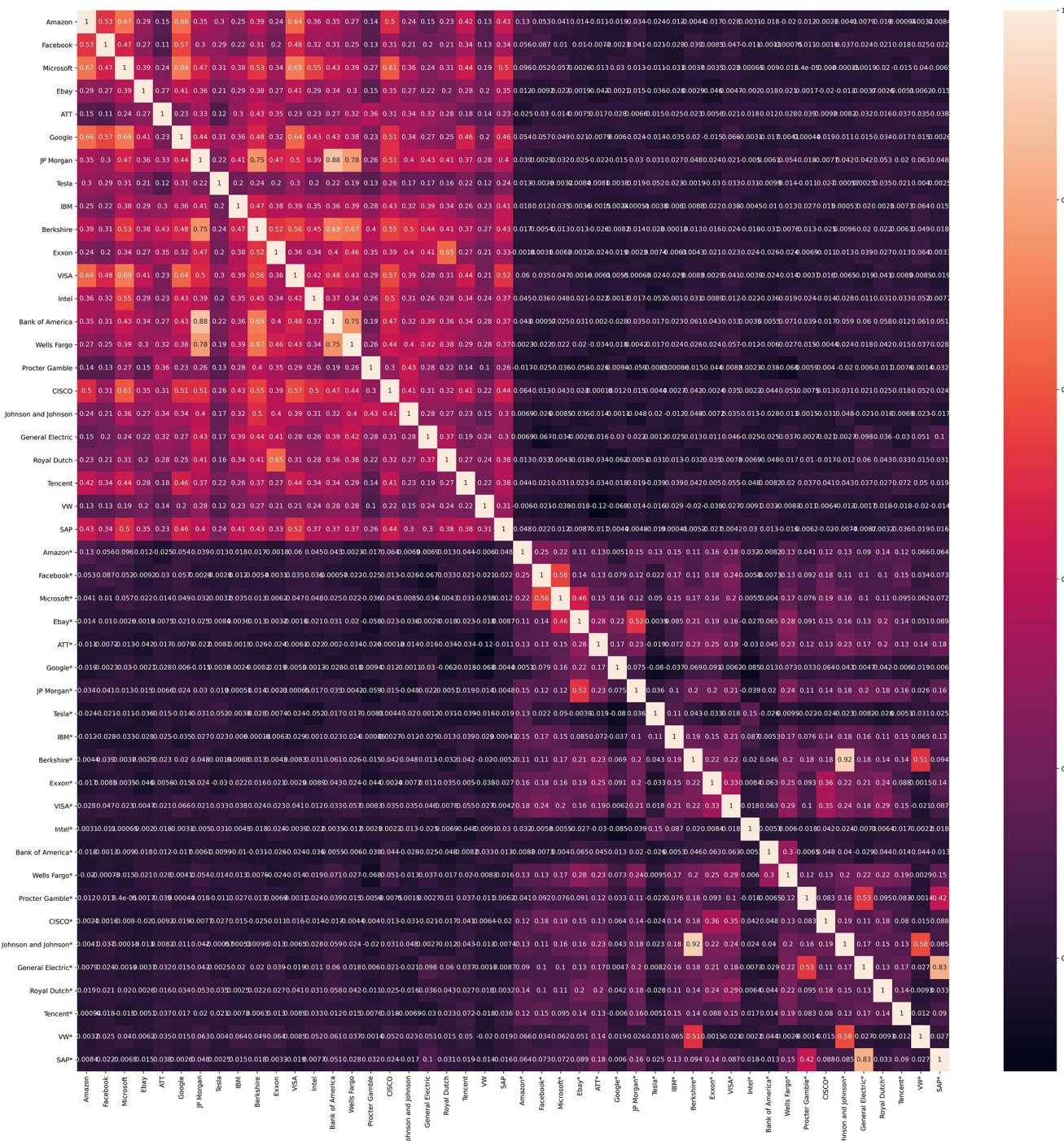

**Fig 3. Correlation matrix of the companies and the Sentiment Indexes, it can be appreciated that the companies are considerably correlated and the same for the Sentiment Indexes, yet there is little to no correlation between stocks and Sentiment Indexes.**

different industries and capitalize in different stock markets. We have Tech companies, Oil and Gas, Banking, Software and Automobile industries.

The Effective Transfer Entropy was calculated as well and presented in the Tables 2–4. In Table 2, the information flux is 67% stronger in the direction of the stock market ($Y \rightarrow X$). The

**Table 2. Information flux from $Y \rightarrow X$ TE.**

| | Shannon Transfer Entropy Results: | | | | | | |
|---|---|---|---|---|---|---|---|
| | $X \rightarrow Y$ TE | | | $Y \rightarrow X$ TE | | | |
| Company | TE | Eff. TE | p-values: | TE | Eff. TE | p-values: | Intensity |
| Amazon | 0.0369 | 0.0161 | 0 | 0.0503 | 0.0211 | 0 | 0.5672 |
| Facebook | 0.0232 | 0.0034 | 0.1967 | 0.0295 | 0.0036 | 0.1 | 0.5142 |
| JP Morgan | 0.0248 | 0.0054 | 0.19 | 0.0454 | 0.0209 | 0 | 0.7946 |
| Tesla | 0.0268 | 0.0047 | 0.1367 | 0.0411 | 0.02 | 0.0033 | 0.8097 |
| IBM | 0.0209 | 0.0027 | 0.1733 | 0.0284 | 0.0078 | 0.06 | 0.7428 |
| Berkshire | 0.0251 | 0.0036 | 0.1033 | 0.0342 | 0.008 | 0.0367 | 0.6896 |
| Exxon | 0.0249 | 0.0047 | 0.13 | 0.0437 | 0.0191 | 0 | 0.8025 |
| VISA | 0.0458 | 0.012 | 0.04 | 0.0562 | 0.0186 | 0 | 0.6078 |
| Wells Fargo | 0.0284 | 0.0075 | 0.07 | 0.0308 | 0.0087 | 0.0067 | 0.537 |
| Royal Dutch | 0.0145 | 0.0017 | 0.21 | 0.0231 | 0.0064 | 0.06 | 0.7901 |
| Ten Cent | 0.0366 | 0.0135 | 0.02 | 0.0479 | 0.0182 | 0.0033 | 0.5741 |
| Volkswagen | 0.0266 | 0.0031 | 0.19 | 0.0273 | 0.007 | 0.0533 | 0.693 |

**Table 3. Information flux from $X \rightarrow Y$ TE.**

| | Shannon Transfer Entropy Results: | | | | | | |
|---|---|---|---|---|---|---|---|
| | $X \rightarrow Y$ TE | | | $Y \rightarrow X$ TE | | | |
| Company | TE | Eff. TE | p-values: | TE | Eff. TE | p-values: | Intensity |
| AT&T | 0.0343 | 0.0095 | 0.0633 | 0.021 | 0 | 0.3433 | -1 |
| Intel | 0.0487 | 0.0239 | 0 | 0.0323 | 0.0071 | 0.0467 | -0.2290 |
| Johnson& Johnson | 0.0291 | 0.0062 | 0.0633 | 0.0261 | 0.0031 | 0.1233 | -0.3333 |
| General Electric | 0.0331 | 0.0106 | 0.0067 | 0.0317 | 0.0086 | 0.0033 | -0.4479 |
| SAP | 0.0265 | 0.0058 | 0.0567 | 0.0197 | 0 | 0.5967 | -1 |

strongest signals where for Tesla (Ticker \$TSLA) and Exxon (Ticker \$XOM) with an 80% information flux. As in Table 3, the information flux is 60% stronger in the direction of the Sentiment Index($X \rightarrow Y$) in average. The strongest signals are AT&T and SAP, both companies with 100% of the signal flux towards the Sentiment Index. Finally, the last six companies tested back with no statistical conclusive results for Shannon TE. Meaning, that the p value is greater than 0.10 in any direction. Results are presented in Table 4.

The most widely known method for measuring information flux between time series is Granger Causality [45]. This method has its limitations towards confirming information flow,

**Table 4. Non existent information flux.**

| | Shannon Transfer Entropy Results: | | | | | | |
|---|---|---|---|---|---|---|---|
| | $X \rightarrow Y$ TE | | | $Y \rightarrow X$ TE | | | |
| Company | TE | Eff. TE | p-values: | TE | Eff. TE | p-values: | Intensity |
| Microsoft | 0.0169 | 0 | 0.63 | 0.0188 | 0 | 0.7933 | NA |
| Ebay | 0.0183 | 0 | 0.5933 | 0.0241 | 0.0027 | 0.1467 | NA |
| Google | 0.0187 | 0 | 0.78 | 0.0209 | 0 | 0.6067 | NA |
| Bank Of America | 0.0231 | 0.0045 | 0.24 | 0.0277 | 0.004 | 0.11 | NA |
| Procter & Gamble | 0.0297 | 0.0031 | 0.22 | 0.0212 | 0 | 0.45 | NA |
| CISCO | 0.0213 | 0 | 0.4567 | 0.0297 | 0.0037 | 0.1367 | NA |

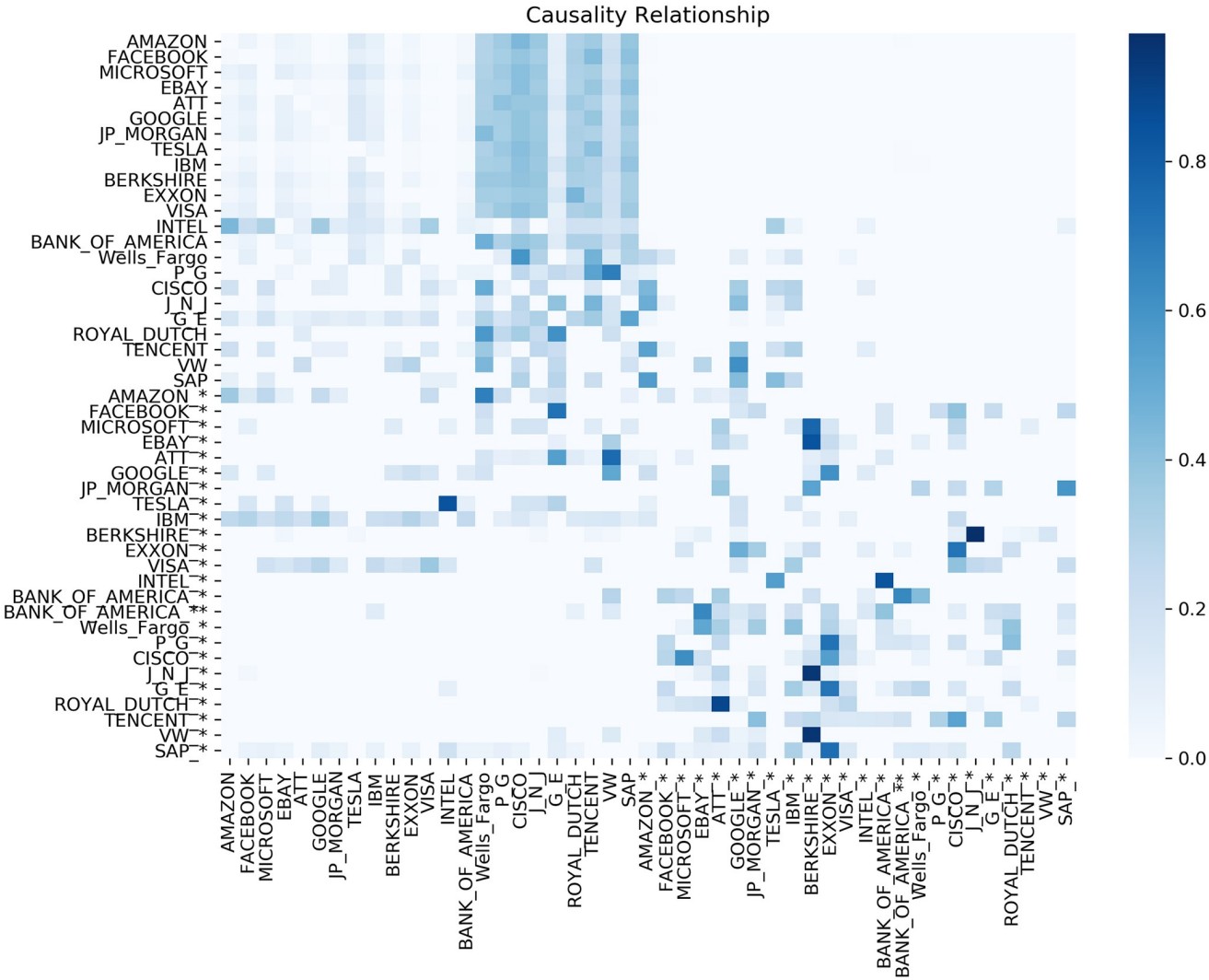

**Fig 4. Granger Causality relationship with a lag of 3 and p-value $p \leq 0.10$.**

compared with Transfer Entropy (TE) [30], which considers mutual information and dynamics transports. Schreiber concludes in his paper that TE is capable of detecting the direction and intensity of signal exchange between two systems whilst ignoring static correlations. A virtue of the TE method is that it allows quantifying information transfer without being bounded by linear dynamics [30].

In Fig 4 the results of Granger Causality tests are presented. In Fig 5 we present the results of the Shannon ETE for the combination of all of our variables, combining stock price returns and Sentiment Indexes. Both results are very similar with a few minor variations. For example, it is visible in Fig fig:Grangercausality some variables with high communication such as Ebay* towards VW and BRKA* receiving signals from MSFT*, EBAY*, JnJ* and VW*. Comparing the same variables results in Fig 5 it is restated that for ETE, EBAY* sends a signal to VW, while BRKA* sends signals to MSFT*, JnJ* and WV.

Comparing ETE Fig 5 with Correlation matrix Fig 3 the main differences are noticeable, the increased signal activity from the stock market to the indexes, and the signal directions.

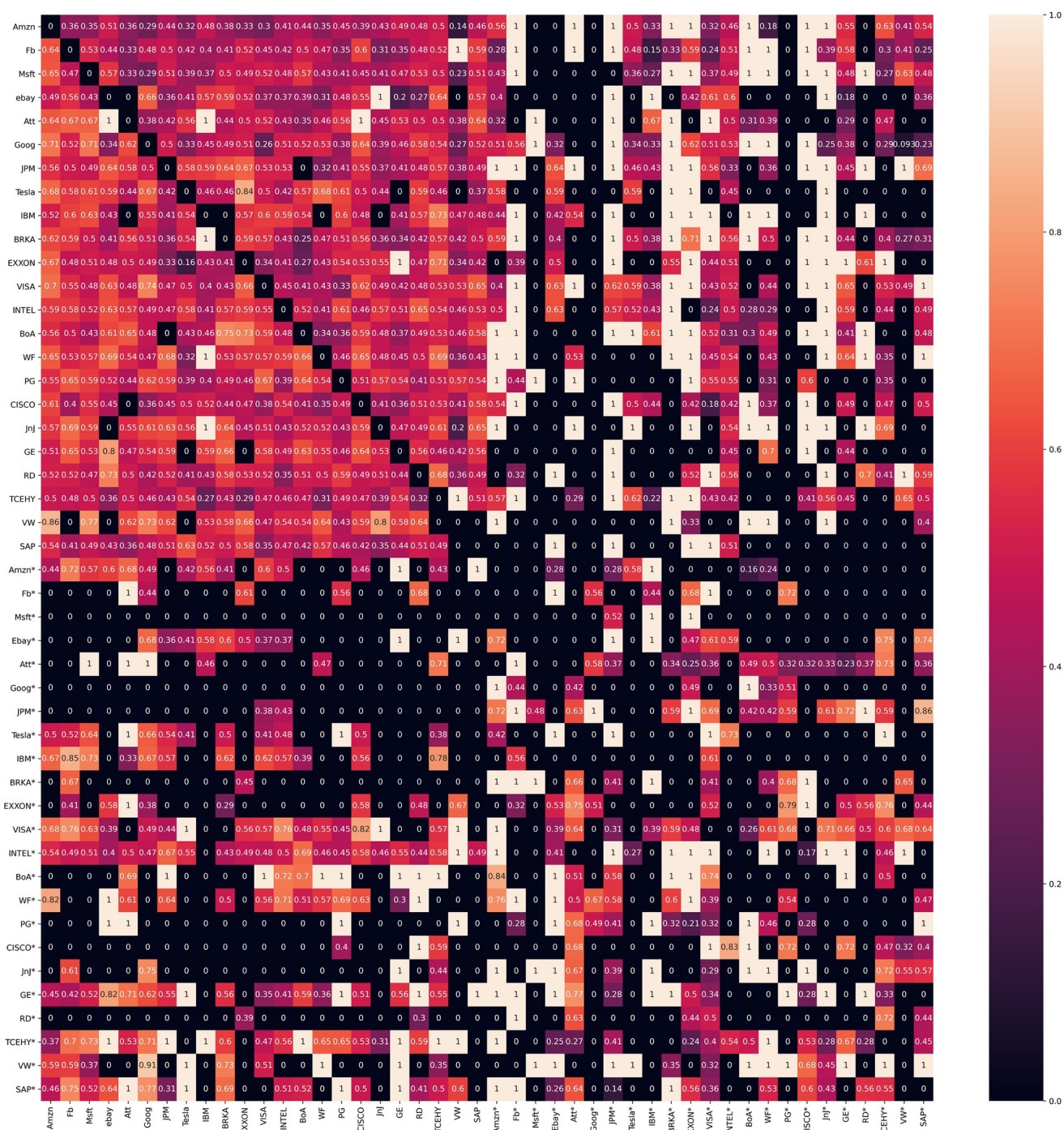

**Fig 5. Shannon ETE matrix with a lag of 3 and p-value $p \leq 0.10$.** Contrary to the correlation matrix there is information transfer between stock companies and the Sentiment Index.

From stock price to stock price we can observe the communication between our variables very similar to the correlation matrix, yet in the ETE column, the direction of the signal is conspicuous. Being the main difference in the Stock market/Stock indexes quadrants, where the correlation is virtually nonexistent, the ETE signal is very present.

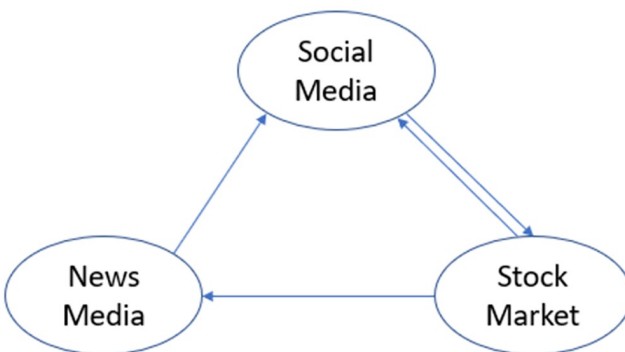

**Fig 6. Theoretical information flux in which the results from Table 2 demonstrate information flow from social media towards stock price and in some cases information flow from stock market towards social media.**

## Discussion

Some studies present evidence of increased drift in stock behaviour after news release of a particular company in a short period of time [46], and that the effect is greater when negative news is released. The information flux measurement between news and financial performance of public traded entities has been proposed [47]. Compared with other computational methods, Transfer Entropy has demonstrated better results in measuring the influence of news in stock returns behavior.

In order to understand the information flux between stock markets and general public opinion, a theoretical framework is presented in Fig 6. In this framework, it is stated that the stock market sends signals to both news media and social networks. In addition, the social network also receives information from news media, sending signals to the stock market in consequence. In Table 2 it is visible how most companies only receive information from social media, while Amazon, Visa, Wells Fargo and Ten Cent receive and send signals to social media. It has been acknowledged throughout the presented investigation that tweets are receiving the information from News Media since some of them quote the source from news companies and others are just retweeting the information from verified news media accounts.

The presented theoretical framework is similar to the Tetlock statement, in which investors obtain their information from secondary data sources. An expansion to Tetlock's model is proposed by considering the signal feedback from investors and general opinion to stock market performance.

It can be stated from the three analyzed methods, that the stocks are highly dependent on their Sentiment Indexes and in some cases from Sentiment Indexes of other companies, meaning that the ecosystem in which the companies operate, even when some companies are in different markets they are affected and affect each other.

The results of this study compared to the results from Bollen et al. [23] differ mainly in the improved text processing algorithm. The second difference, is that our study extend the analysis to 23 different companies, some of them that operate in non-English speaking countries with positive results.

This model obtained positive results considering that our sample covered 6 years with a volume of 200,000 tweets for 23 companies, contrasted to a previous study [48] that used a sample of 60,000 tweets for a 6 day period of time. In which authors concluded that the sample size prevented their study from acquiring statistical significant results using correlation and regression tests. While both studies conclude that correlations between stock prices and sentiment

indexes are nearly non-existent, Transfer Entropy definitely demonstrates statistical significance between Social Media behaviour and Stock price reaction.

## Conclusion

In this paper the relationship between stock price performances and social sentiment in public social networks was reviewed, i.e. there is enough evidence to support that what investors and observers of stock markets affect the performance of it independently of the level of participation or location of both the market and the participants issuing the opinion. The database covers a conveniently long period of time (2013–2018) after the 2008 crisis until before the economic slowdown of 2019. The opinions data set was considerably large (over 200,000 tweets covering the same period of time).

By proposing an original simplified Sentiment Index using technological tools as web scraping and machine learning for NLP we evaluate the relationship between variables that apparently are not compatible firsthand. Our simulations were performed for each company and corresponding Sentiment Index. In addition, We have used different statistical methods to measure the performance of our index construction model, in which the results proved to be an improved model over existing methods. It is important to mention that throughout the literature reviewed, it is noticeable that past studies have been consistent in comparing the sentiment analysis versus a single stock. However, they haven't studied the indirect bounce of data among a similar range of stocks. As it is to be proven, there is undoubtedly communication between multiple Sentiment Indexes affecting multiple companies; meaning that the comments regarding a particular company affect indirectly the performance of several others.

The asymmetric theory of information states that negative news has a greater impact on stock prices than positive news. In this study, we present visual evidence to raise the question, expand research aimed to support this theory. Future work can be focused on creating an independent index for both positive and negative news/opinions and measure the marginal effects of each index in the stock price. There is also ground for replicating the model, measuring the extent of the network effects with other exogenous variables in the stock market.

## Supporting information

**S1 Data.**
(ZIP)

## Acknowledgments

We thank to M. Sc. José G. Mendoza Macias who provided computer equipment, Professor Mendoza also served as link with the Mexican Artificial Intelligence Society and introduced us to the top researchers of financial institutions in Latin America for a better understanding of the applications of modern algorithms in financial sciences.

We thank to Dr. José Ernesto Amorós (EGADE Business School), headmaster of the Ph.D department.

## Author Contributions

**Conceptualization:** José Antonio Nuñez Mora.

**Formal analysis:** José Antonio Nuñez Mora.

**Investigation:** Andrés García-Medina.

**Methodology:** Andrés García-Medina.

**Supervision:** Andrés García-Medina.

**Validation:** Andrés García-Medina, José Antonio Nuñez Mora.

**Writing – original draft:** Román Alejandro Mendoza Urdiales.

**Writing – review & editing:** Román Alejandro Mendoza Urdiales, Andrés García-Medina.

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
