## [Decision Letter · Decision Letter 0]

13 May 2021

PONE-D-21-09517

Measuring information flux between social media and stock prices with Transfer Entropy

PLOS ONE

Dear Dr. Mendoza Urdiales,

Thank you for submitting your manuscript to PLOS ONE. After careful consideration, I feel that it has merit but does not fully meet PLOS ONE’s publication criteria as it currently stands. 

I find this paper interesting and suitable of publication in Plos One. However, reviewers have some concerns about methodological aspects of the research that need to be addressed in a major revision.

I will suggest to the authors to compare their findings with the ones obtained by other authors. This will help to show the contribution of this research, which is not clear in my opinion. In the introduction section, authors write about the EMF, the MDH and the SIAH. At this point, I will suggest commenting main aspects of the Adaptative Market Hypothesis and the Fractal Market Hypothesis.

Other minor questions are for example when authors use reference numbers, as in line 25, the name of the author must be before. Please revise this in the manuscript.

I invite you to submit a revised version of the manuscript that addresses the points raised during the review process.

We look forward to receiving your revised manuscript.

Kind regards,

J E. Trinidad Segovia

Academic Editor

PLOS ONE

Journal Requirements:

2.In your Data Availability statement, you have not specified where the minimal data set underlying the results described in your manuscript can be found. PLOS defines a study's minimal data set as the underlying data used to reach the conclusions drawn in the manuscript and any additional data required to replicate the reported study findings in their entirety. All PLOS journals require that the minimal data set be made fully available. For more information about our data policy, please see http://journals.plos.org/plosone/s/data-availability.

Reviewers' comments:

Reviewer's Responses to Questions

**Comments to the Author**

1. Is the manuscript technically sound, and do the data support the conclusions?

Reviewer #1: Yes

Reviewer #2: Yes

2. Has the statistical analysis been performed appropriately and rigorously? 

Reviewer #1: Yes

Reviewer #2: Yes

3. Have the authors made all data underlying the findings in their manuscript fully available?

Reviewer #1: No

Reviewer #2: Yes

4. Is the manuscript presented in an intelligible fashion and written in standard English?

Reviewer #1: No

Reviewer #2: Yes

5. Review Comments to the Author

Reviewer #1: The manuscript by Mendoza Urdiales, García-Medina and Nuñez Mora presents an interesting study of the flux between information in social media and stock prices. For this purpose, the authors have analysed over 200.000 tweets from a six years period and also the evolution of stock prices of major companies from different markets. These two data sets have been correlated using the transfer entropy, as well as the Pearson correlation and other entropy measures. A positive noticeable flow of information between social media (at least this platform) and stock prices is reported and quantified. The paper is well written, the objective is well presented and the conclusions are clear. I am therefore in favour of providing a positive recommendation. However, some parts of the paper are poorly written and it is difficult to follow for non-specialists, thus I recommend that the authors improve these parts before the paper is accepted.

In particular, the following issues should be solved:

- How is the polarity defined and calculated? Apparently, it is only the sum of the polarities of the words in a sentence. Is that so? The meaning can change notably by changing the ordering of the words. Are some specific constructions recognized?

- Please check the definition of Y_t. Currently its states "For a given day t, the Polarity of the Day Y was constructed by adding ...". Probably Y refers to the company. Similarly, improve the definition of X_t.

- Equation (5) is the core of the analysis performed. Thus it should be clearly explained. What are the indices k and l? Also, what are x_t^{(k)} and y_t^{(l)}? How are the transition probabilities calculated?

- The comment in the first paragraph in pg. 8 is unclear. The data (both from Twitter and stock prices) are sampled daily, so that a discrete set is obtained. Why is this a problem for the application of Eq. (5) which also refers to discrete datasets?

- The authors report a negative Pearson correlation coefficient between companies and indexes (last line of the first paragraph in pg. 9). Where is this shown? One cannot notice any red spots in Fig. 3.

- In tables 2, 3 and 4, a p value is given. How is this calculated? What does it imply? It is not mentioned in the "Methodology section"

- What evidence is given to support that the negative news have a greater impact in prices than positive ones? This goes unnoticed in the main text, but concluded.

- The use of English must also be improved in some parts.

Reviewer #2: In the article, the authors offer a contribution to answering the question that, as well expressed by them, is “How can we translate this impact in a quantitative method and in a statistically measurable fashion that can be scientifically replicated?”

A very thorough introduction substantiates the claim of the authors that there is an important relation between news, or the comments of experts, on the way stock prices move.

The way the authors describe the Language Processing algorithm is very enlightening, and results were obtained in a series of different ways, making them more robust.

Figure 5 is THE figure with the major explanation capability and impact in the article.

A suggestion for future work would be doing this in moving windows, so as to add a dimension of time to the analysis being made.

There are some minor issues I would like to be addressed by the authors.

1. I think some of the captions should be enlarge. A caption should be understood without the need to resort to the main text. Particularly, for Figure 1, Figure 3.

2. In equation (3), the use of the difference between stock prices and not the return, which would be divided by the price of the day before, does not introduce problems with stationarity of the time series? This is issue demands some explanation, as it will be questioned by many readers.

3. Why are the authors using k=l=3 for Transfer Entropy? Maybe it would be good to elaborate on it.

4. In the Conclusion, the authors write: “The asymmetric theory of information states that the negative news have a greater impact in stock prices than positive news. In this study we present evidence to support this theory.” It was nota t all clear to me where the results that were presented substantiate this importante claim, as I haven’t seen a separate analysis for positive and negative sentiments.

Suggestions for grammar or orthography.

Please change, at the author’s discretion:

• “Farag and Cressy [29], studied how price limits, aimed to prevent speculation 11 amongst traders when new information is released in the market.” The paragraph is nuclear.

• “in a random matter” to “in a random manner”.

• “according the information” to “according to when the information”.

• “market movements that coincides” to “market movements that coincide”.

• “of 2018. Which” to “of 2018, which”.

• “etc), this step” to “etc); this step”.

• Please correct punctuation in displayed equations.

• “Even so, provide” to “Even so, it provides”.

• “statistical analysis the combination” to “statistical analysis of the combination”.

• “For measuring the signal transfer” to “In order to measure the signal transfer”.

• “in different stock markets, we” to “in different stock markets. We”.

• “information flow. Compared” to “information flow, compared”.

• There are plenty others. I recommend a thorough review on writing.

6. PLOS authors have the option to publish the peer review history of their article (what does this mean?). If published, this will include your full peer review and any attached files.

Reviewer #1: No

Reviewer #2: No

---

## [Author Response · Author response to Decision Letter 0]

30 Jun 2021

The finding were compared with ones obtained by other authors and expresed in the Discussion and Conclusion sections of the paper, emphazising the information flow between multiple indexes and companies contrasted to most literature reviewed that only measure information flow between paired stock prices and their respective sentiment index. In the introduction section,main aspects of the Adaptative Market Hypothesis and the Fractal Market Hypothesis were added as suggested.

The Databases were added to the file and all the remaining suggestions attended:

 1) Reviewer #1’s observation: ”- How is the polarity defined and calculated? Apparently, it is only the sum of the polarities of the words in a sentence. Is that so? The meaning can change notably by changing the ordering of the words. Are some specific constructions recognized?”

Response: In lines 198-214 we expand the definition of the algorithm and how its logic is structured, considering the inefficiency when words are not correctly ordered. The source code was added into the 3rd footnote at the end of page 7.

2) Reviewer #1’s observation: “- Please check the definition of Y_t. Currently its states "For a given day t, the Polarity of the Day Y was constructed by adding ...". Probably Y refers to the company. Similarly, improve the definition of X_t.”

Response: In lines 248-255 were corrected and improved according to the feedback from the reviewer. 

3) Reviewer #1’s observation: “- Equation (5) is the core of the analysis performed. Thus, it should be clearly explained. What are the indices k and l? Also, what are x_t^{(k)} and y_t^{(l)}? How are the transition probabilities calculated?”

Response: In lines 271-275 we introduced the definitions of k and l, furthermore the explanation of why the combinations of k=l=3 was chosen for presenting the results were introduced in lines 359-363.

4) Reviewer #1’s observation: “- The comment in the first paragraph in pg. 8 is unclear. The data (both from Twitter and stock prices) are sampled daily, so that a discrete set is obtained. Why is this a problem for the application of Eq. (5) which also refers to discrete datasets?”

Response: In lines 296-300 we introduce the configuration of the model to data discretization in which a number of bins is selected in order to partition the data, since the time series X and Y have the characteristic of being continuous. In lines 358-359 the quantiles selected to discretize the data was presented.

5) Reviewer #1’s observation: “- The authors report a negative Pearson correlation coefficient between companies and indexes (last line of the first paragraph in pg. 9). Where is this shown? One cannot notice any red spots in Fig. 3.”

Response: In lines 307-308 we correct the interpretation of the graph in which there’s little to no correlation between the stock prices and indexes. Furthermore, in the Figure 3 the explanation was improved by mentioning this correct interpretation.

6) Reviewer #1’s observation: “- In tables 2, 3 and 4, a p value is given. How is this calculated? What does it imply? It is not mentioned in the "Methodology section"

Response: In lines 321-324, it is introduced how the p -value is calculated for the null hypothesis for no information exchange for the simulated distribution that corresponds for the transfer entropy estimations.

7) Reviewer #1’s observation: “- What evidence is given to support that the negative news has a greater impact in prices than positive ones? This goes unnoticed in the main text but concluded.”

Response: The first comment regarding negative news with greater impact that positive news is introduced in line 59. Further evidence is introduced in line 418 and presented Figure 2 and Figure 7 which visually helps with the interpretation that there is concordance with the cited article “Wesley S. Chan. Stock price reaction to news and no-news: drift and reversal after headlines. Journal of Financial Economics. 2001; 18:203–260” . There is also mentioned that when the companies that presented information flow from Sentiment Index towards the Stock Price are plotted in parallel it can be appreciated the aforementioned effect. 

In line 445 the conclusion was adjusted to present a precedent for further investigation given the evidence presented in the introduction and explained further in the Discussion section.

Reviewer #2: In the article, the authors offer a contribution to answering the question that, as well expressed by them, is “How can we translate this impact in a quantitative method and in a statistically measurable fashion that can be scientifically replicated?”

A very thorough introduction substantiates the claim of the authors that there is an important relation between news, or the comments of experts, on the way stock prices move. The way the authors describe the Language Processing algorithm is very enlightening, and results were obtained in a series of different ways, making them more robust.

Figure 5 is THE figure with the major explanation capability and impact in the article.

A suggestion for future work would be doing this in moving windows, so as to add a dimension of time to the analysis being made.

1) Reviewer #2’s observation: “1. I think some of the captions should be enlarge. A caption should be understood without the need to resort to the main text. Particularly, for Figure 1, Figure 3.”

Response: Figure’s captions were enlarged according to the authors’ suggestion, in which in Figure 1 we explain the steps involved in the generalized framework followed to get to our model results. In Figure 3 we improved the correlation matrix graph and corrected the explanation.

2) Reviewer #2’s observation: “In equation (3), the use of the difference between stock prices and not the return, which would be divided by the price of the day before, does not introduce problems with stationarity of the time series? This is issue demands some explanation, as it will be questioned by many readers.”

Response: In line 368 we introduce the interest in working with stationary version of the Stock price, this to change most of the literature review that prefer to work with the stock performance. To demonstrate that it would not present issues, stationary tests were performed as explained in lines 270-375 in which all tests confirmed that the time series satisfied the statistical requirements.

3) Reviewer #2’s observation: “Why are the authors using k=l=3 for Transfer Entropy? Maybe it would be good to elaborate on it.”

Response: The definition of how k and l are calculated are introduced in lines 271-275 and equation 6 was added, in lines 364-368 we explain why k=l=3 was selected and what would be the interpretation in the results. Meaning that the impact of the sentiment index would have its maximum effect in a 3-day span.

4) Reviewer #2’s observation: “In the Conclusion, the authors write: “The asymmetric theory of information states that the negative news have a greater impact in stock prices than positive news. In this study we present evidence to support this theory.” It was not at all clear to me where the results that were presented substantiate this important claim, as I haven’t seen a separate analysis for positive and negative sentiments.”

Response: The first comment regarding negative news with greater impact that positive news is introduced in line 59. Further evidence is introduced in line 418 and presented Figure 2 and Figure 7 which visually helps with the interpretation that there is concordance with the cited article “Wesley S. Chan. Stock price reaction to news and no-news: drift and reversal after headlines. Journal of Financial Economics. 2001; 18:203–260” . There is also mentioned that when the companies that presented information flow from Sentiment Index towards the Stock Price are plotted in parallel it can be appreciated the aforementioned effect. 

In line 445 the conclusion was adjusted to present a precedent for further investigation given the evidence presented in the introduction and explained further in the Discussion section.

5) Reviewer #2’s observation: Suggestions for grammar or orthography.

 Please change, at the author’s discretion:

 “Farag and Cressy [29], studied how price limits, aimed to prevent speculation 11 amongst traders when new information is released in the market.” The paragraph is nuclear.

• “in a random matter” to “in a random manner”.

• “according the information” to “according to when the information”.

• “market movements that coincides” to “market movements that coincide”.

• “of 2018. Which” to “of 2018, which”.

• “etc), this step” to “etc); this step”.

• Please correct punctuation in displayed equations.

• “Even so, provide” to “Even so, it provides”.

• “statistical analysis the combination” to “statistical analysis of the combination”.

• “For measuring the signal transfer” to “In order to measure the signal transfer”.

• “in different stock markets, we” to “in different stock markets. We”.

• “information flow. Compared” to “ ”.

• There are plenty others. I recommend a thorough review on writing.

Response: The grammar was reviewed according as requested

5) Reviewer #2’s observation: “I will suggest to the authors to compare their findings with the ones obtained by other authors. This will help to show the contribution of this research, which is not clear in my opinion. In the introduction section, authors write about the EMF, the MDH and the SIAH. At this point, I will suggest commenting on the main aspects of the Adaptive Market Hypothesis and the Fractal Market Hypothesis.”

Response: Comparisons to other 2 models were presented in lines 440-452. In addition, it was expressed that the major contribution from this research is the comparison of a company’s index versus other companies and that there is communication between multiple companies affected by multiple Sentiment Indexes in lines 465-470. In the introduction the AMH and FMH were added as suggested, in lines 11-32.

---

## [Decision Letter · Decision Letter 1]

21 Jul 2021

PONE-D-21-09517R1

Measuring information flux between social media and stock prices with Transfer Entropy

PLOS ONE

Dear Dr. Mendoza Urdiales,

Thank you for submitting your manuscript to PLOS ONE. After careful consideration, I feel that it has merit but does not fully meet PLOS ONE’s publication criteria as it currently stands. 

Major concerns have been properly addressed in this new version. However, still some doubts persist. One of the reviewers consider that the methodology remains incomplete. Therefore, I invite you to submit a revised version of the manuscript that addresses the points raised during the review process.

We look forward to receiving your revised manuscript.

Kind regards,

J E. Trinidad Segovia

Academic Editor

PLOS ONE

Journal Requirements:

Reviewers' comments:

Reviewer's Responses to Questions

**Comments to the Author**

1. If the authors have adequately addressed your comments raised in a previous round of review and you feel that this manuscript is now acceptable for publication, you may indicate that here to bypass the “Comments to the Author” section, enter your conflict of interest statement in the “Confidential to Editor” section, and submit your "Accept" recommendation.

Reviewer #1: All comments have been addressed

Reviewer #2: All comments have been addressed

2. Is the manuscript technically sound, and do the data support the conclusions?

Reviewer #1: Partly

Reviewer #2: Yes

3. Has the statistical analysis been performed appropriately and rigorously? 

Reviewer #1: Yes

Reviewer #2: Yes

4. Have the authors made all data underlying the findings in their manuscript fully available?

Reviewer #1: Yes

Reviewer #2: Yes

5. Is the manuscript presented in an intelligible fashion and written in standard English?

Reviewer #1: Yes

Reviewer #2: Yes

6. Review Comments to the Author

Reviewer #1: In the new version of the manuscript, the authors have amended some sections, and improved the overall quality. The rest of points in my previous report have been addressed in the respond letter.

In my opinion, the authors have improved the manuscript but new questions have rised. More importantly, the description of their methodology is still incomplete. In some parts it is difficult to follow their reasoning and I doubt one can replicate their methodology following the explanations. For instance, the meaning of l and k in eq. (5) is still unexplained. The new sentence in lines 360-361 is unclear. What is c(5,95)? The following paragraph is also confusing (probably due to a grammatical error).

I also, I would the authors to take into consideration the following points:

- When the intensity of the signal is negative, are the same values of l and k used? One would expect a different mechanism there, and therefore a different delay.

- Fig. 6 does not show a clear cut proof that negative signals have a stronger effect on the price than positive ones. This conclusion is therefore not sufficiently supported and should be weakened, or further analysis must be made.

Despite these comments, my general opinion of the manuscript is positive and I recommend the paper for publication.

Reviewer #2: All comments of both reviewers have been addressed. I think the article is now ready to be published in Plos One.

7. PLOS authors have the option to publish the peer review history of their article (what does this mean?). If published, this will include your full peer review and any attached files.

Reviewer #1: No

Reviewer #2: No

---

## [Author Response · Author response to Decision Letter 1]

16 Aug 2021

List of responses:

For instance, the meaning of l and k in eq. (5) is still unexplained. The new sentence in lines 360-361 is unclear. What is c(5,95)? The following paragraph is also confusing (probably due to a grammatical error).

R: Definitions of l and k were extended in lines 262-277. The lines 360-365 were improved and explanation was extended. The definition of c(5,95) was also considered in the paragraph

I also, I would the authors to take into consideration the following points:

- When the intensity of the signal is negative, are the same values of l and k used? One would expect a different mechanism there, and therefore a different delay.

R: Regarding the combination of lags mentioned, we performed simulations with all possible combinations (with k=1,2,3; l=1,2,3, for X and Y respectively. Where k=l=3, the quantity of results within our sample with statistical significancy and were the ones presented in the paper. Information transference from both X->Y and Y->X (p-value=<0.1) was the largest in any of the scenarios:

a) ETE X->Y with p-value =<0.10 and ETE Y->X with p-value >0.10 , meaning that there is only information transfer from X to Y. 

b) ETE X->Y with p-value >0.10 and ETE Y->X with p-value =<0.10 , meaning that there is only information transfer from Y to X. 

c) ETE X->Y with p-value =<0.10 and ETE Y->X with p-value =<0.10 , meaning that there is information flux from both variables X to Y and viceversa. 

To differentiate information flow, X of Y, in Eq. (9), we defined how the intensity of the signal would be measured:

In this equation, if the information signal was greater from Y-> X, it would take a positive sign, meaning that there is a positive signal flowing from social media to stock prices, which is the main interest of our study. The contrary sign could be expected if the signal from X->Y was greater. This was performed exclusively to differentiate the direction of the signal in our table results. 

In addition, that the effect of the news in the stock market 3 days after the news is released is a result confirmed that well other authors shared in different papers quoted (Johan Bollen, Huina Mao, Xiaojun Zeng. Twitter mood predicts the stockmarket. Journal of Computational Science.2011; 2:1–

Tahir M. Nisar, Man Yeung. Twitter as a tool for forecasting stock marketmovements: A short-window event study. The Journal of Finance and DataScience 4.2018:101-119)

- Fig. 6 does not show a clear-cut proof that negative signals have a stronger effect on the price than positive ones. This conclusion is therefore not sufficiently supported and should be weakened, or further analysis must be made.

Figure 6 was omitted for this review to avoid ambiguity in the presented results

---

## [Decision Letter · Decision Letter 2]

8 Sep 2021

Measuring information flux between social media and stock prices with Transfer Entropy

PONE-D-21-09517R2

Dear Dr. Mendoza Urdiales,

We’re pleased to inform you that your manuscript has been judged scientifically suitable for publication and will be formally accepted for publication once it meets all outstanding technical requirements.

Kind regards,

J E. Trinidad Segovia

Section Editor

PLOS ONE

Additional Editor Comments (optional):

Reviewers' comments:

Reviewer's Responses to Questions

**Comments to the Author**

1. If the authors have adequately addressed your comments raised in a previous round of review and you feel that this manuscript is now acceptable for publication, you may indicate that here to bypass the “Comments to the Author” section, enter your conflict of interest statement in the “Confidential to Editor” section, and submit your "Accept" recommendation.

Reviewer #1: All comments have been addressed

2. Is the manuscript technically sound, and do the data support the conclusions?

Reviewer #1: Yes

3. Has the statistical analysis been performed appropriately and rigorously? 

Reviewer #1: Yes

4. Have the authors made all data underlying the findings in their manuscript fully available?

Reviewer #1: Yes

5. Is the manuscript presented in an intelligible fashion and written in standard English?

Reviewer #1: Yes

6. Review Comments to the Author

Reviewer #1: The authors have considered all the points in my previous report and have modified the manuscript accordingly. I am happy to recommend publication of the manuscript.

7. PLOS authors have the option to publish the peer review history of their article (what does this mean?). If published, this will include your full peer review and any attached files.

Reviewer #1: No

---

## [Editor Report · Acceptance letter]

13 Sep 2021

PONE-D-21-09517R2 

Measuring information flux between social media and stock prices with Transfer Entropy 

Dear Dr. Mendoza Urdiales:

I'm pleased to inform you that your manuscript has been deemed suitable for publication in PLOS ONE. Congratulations! Your manuscript is now with our production department. 

Kind regards, 

on behalf of

Dr. J E. Trinidad Segovia 

Section Editor

PLOS ONE